# Post-Stroke Bacteriuria: A Longitudinal Study among Stroke Outpatients and Inpatients at the Korle-Bu Teaching Hospital in Ghana

**DOI:** 10.3390/medsci5020011

**Published:** 2017-06-02

**Authors:** Eric S. Donkor, Samuel Darkwah, Albert Akpalu

**Affiliations:** 1Department of Medical Microbiology, School of Biomedical and Allied Health Sciences University of Ghana, Accra, Ghana; kwekuadarkwah@gmail.com; 2Department of Medicine, School of Medicine and Dentistry, University of Ghana, Accra, Ghana; a_akpalu@yahoo.com

**Keywords:** stroke, urinary tract infection, inpatient, Ghana, *E. coli*

## Abstract

Infections of the urinary tract constitute an important post-stroke complication but in Africa, little is known about such infections in relation to stroke patients. The aim of the study was to investigate the epidemiology of bacteriuria among stroke patients at the Korle-Bu Teaching Hospital (KBTH) in Ghana including the prevalence, incidence, risk factors and aetiological agents. This was a longitudinal study involving 55 outpatients and 16 inpatients of stroke from the physiotherapy clinic and stroke admission ward of KBTH respectively. Urine cultures for inpatient subjects were done each day until the patient was discharged. With outpatients, urine specimens were analysed every week or two for 6 months. Information on demographics and clinical history of the study participants were extracted from their clinical records. The results showed that the prevalence of bacteriuria among stroke outpatients and inpatients were 10.9% (6/55) and 18.8% (3/16) respectively (*p* = 0.411). Among both the outpatients and inpatients, there was one new case of bacteriuria each during the period of follow-up. Overall, 1/9 (11%) of the bacteriuria cases among the stroke patients was symptomatic. Severe stroke (OR = 17.7, *p* = 0.008) and pyuria (OR = 38.7, *p* = 0.002) were identified as predictors of bacteriuria. *Escherichia coli* was the most common organism implicated in bacteriuria and was susceptible to amikacin, but resistant to augmentin, ampicillin, cefuroxime, cotrimoxazole, meropenem, norfloxacin and tetracycline. Overall, bacteriuria is a common complication among both stroke inpatients and outpatients at KBTH, though it appears to be more common among the former. Stroke severity appears to be the main stroke-related determinant of bacteriuria among stroke patients. Bacteriuria among stroke patients is mainly asymptomatic and *E. coli* is the most important aetiological agent.

## 1. Introduction

Stroke is ranked as the second leading cause of death worldwide with an annual mortality rate of 6 million, a toll exceeding that of human immunodeficiency virus/acquired immune deficiency syndrome (HIV/AIDS), malaria and tuberculosis all together [1]. The burden of stroke does not only lie in the high mortality, but the high morbidity also results in up to 50% of survivors being chronically disabled [2]. Thus, stroke is a disease of immense public health importance with economic and social consequences. A few years ago, stroke was considered a disease of the developed world. However, the burden of stroke seems to be shifting to the developing world and currently two-thirds of stroke mortality cases occur in Sub-Saharan Africa [2,3], where poverty, malnutrition and communicable diseases such as HIV/AIDS also exert their greatest toll. In Ghana, stroke ranks among the top three causes of mortality [4]. A one year retrospective study of in-patients with stroke admitted to a teaching hospital in Ghana showed that stroke constituted 9.1% of the total medical adult admissions and 13.2% of all medical adult deaths [5].

Stroke patients are known to have some impairment in immunity and are thus prone to several infections [6]. Bacteriuria is an important complication among stroke patients and it may be classified as urinary tract infection (UTI) or asymptomatic bacteriuria (ASB). While UTI is generally defined in association with genitourinary symptoms and inflammation with a positive urine culture, ASB rather defines the presence of bacteria (>10^5^ cfu/mL) in urine without the evidence of clinical signs and symptoms associated with urinary infections [7,8]. A systematic review reported the prevalence of stroke related UTI to be 10% in the hospital environment, but this review did not include data from developing countries [9]. In healthy individuals, prevalence of UTI vary significantly among different populations and ASB is not uncommon in males and females over 50 years old, and increases with age [8]. UTI is associated with poor stroke outcomes and the risk factors include stroke severity, depressed conscious level, increased post-void residual urine volume and diabetes mellitus [9,10]. Several bacterial pathogens have been implicated in bacteriuria among stroke patients and the major ones include *Staphylococcus aureus* and Gram-negative bacteria such as *Klebsiella pneumoniae*, *Pseudomonas aeruginosa*, *Escherichia coli* and *Enterobacter* spp. [9,10]. The increasing resistance of the causative organisms to antibiotics further heightens the burden of UTI among stroke patients [11,12]. It is thought that prophylactic antibiotics may offer some benefit against UTI in stroke patients. However, the use of such drugs after stroke is still unclear, and questions concerning the risk of selecting resistant strains, defining the best antibiotic regimen, and determining which patients with stroke may benefit from prophylaxis are not fully resolved [13].

In Ghana and other African countries, little is known about bacteriuria in relation to stroke patients. In a recent cross-sectional study, we reported a significantly higher prevalence of bacteriuria among stroke outpatients (24.3%) than among a healthy control group (7.2%) in Ghana [14]. Our data suggests that the burden of post-stroke bacteriuria in Ghana may be several folds higher than what is observed in the developed world [9]. To provide further insights into post-stroke bacteriuria and UTI in Ghana, we carried out a longitudinal study among stroke inpatients and outpatients in the current study. The aim of this study was to investigate the epidemiology of post-stroke bacteriuria among stroke patients at the Korle-Bu Teaching Hospital in Ghana, including the prevalence, incidence, risk factors and aetiological agents.

## 2. Materials and Methods

### 2.1. Study Site

The study was conducted at the Korle-Bu Teaching Hospital (KBTH), in Accra. The hospital is one of the largest hospitals in Africa and the leading national referral centre in Ghana with a bed capacity of about 2000 and 17 clinical and diagnostic departments [15]. The hospital has an average daily attendance of 1500 patients and about 250 patient admissions [15]. There has recently been an establishment of a stroke unit, which admits and takes care of acute stroke cases. This unit has a maximum bed capacity of 15. Korle-Bu Teaching Hospital has a Physiotherapy Department, which provides physical therapeutic and rehabilitative services to both outpatients and inpatients. The department has a clinic and gymnasium that receives and render medical services to outpatients, including stroke patients in need of physiotherapy.

### 2.2. Study Design, Subject Recruitment and Data Collection

This was a longitudinal study involving 71 stroke patients at KBTH from May to October 2015. The study participants comprised 55 stroke outpatients from the physiotherapy clinic of the hospital (patients who had recovered from a past cerebrovascular accident and were receiving physiotherapeutic management to correct physical disability) and 16 stroke inpatients from the stroke ward. Recruitment of the stroke patients was based on definite clinical and/or radiological diagnosis of stroke. Additionally, the stroke outpatients should have had the disease for at least 3 months while the inpatients should have been on admission for not less than 48 h. Patients with ambiguous diagnosis of stroke were excluded from the study. Stroke patients who had been on antibiotics two weeks or shorter before recruitment of the study subjects were also excluded. Eligible study participants were verbally informed of the study and a consent form was provided for formal approval by each study participant. Inpatients who were not in a conscious position to provide consent were consented for by their caregivers.

Information on demographics and clinical history of the study participants were extracted from their clinical records. The demographic information included age, sex, marital status, education and income. The clinical data collected focused on UTI signs/symptoms and various stroke parameters including subtype, duration, severity, frequency, risk factors and side of the body affected. Stroke severity and functional status of both inpatients and outpatients was measured using the modified Rankin’s scale (mRs) described by Bonita et al. [16]. The mRs scores were stratified as mild stroke (1–3) and severe stroke (4–6) [16].

The two groups of study subjects (stroke inpatients and outpatients) were followed up for the development of significant bacteriuria and associated symptoms. Urine sampling and culture from inpatient subjects was attempted each day until the patient was discharged from the stroke ward. With the outpatients, samples were collected at each of the patient’s visit to the physiotherapy clinic every week or two, depending on the patients’ management plan for physiotherapy sessions. The outpatients were followed up for 6 months. None of the study participants were catheterised prior to or during the study. None of the study participants received antibiotics from the study except if it was in line with clinical management of the patient. The Cohort ended for any participant who had to receive antibiotic treatment in the course of clinical management.

### 2.3. Laboratory Analysis

Urine dipsticks were used to perform urinalysis on urine specimens to assess the presence of leucocytes (pyuria) and red cells (haematuria). The urine colour and appearance were observed and recorded. Urine specimens were inoculated onto plates of Blood agar (Oxoid Ltd., Basingstoke, UK), MacConkey agar (Oxoid Ltd., Basingstoke, UK) and Cysteine Lactose Electrolyte Deficient agar (Oxoid Ltd., Basingstoke, UK) using a standard loop calibrated to hold 0.01 mL of urine. The plates were incubated at 37 °C aerobically for 18–24 h. After incubation, bacterial colonies on the agar plates were counted and the results multiplied by the loop volume. A bacterial count of 1 × 10^5^ per mL was considered as significant bacteriuria while counts less than 1 × 10^5^ per mL were considered as no significant bacterial growth [17]. Only samples with significant bacteriuria were considered positive for bacteriuria or UTI. Bacterial isolates were identified based on colonial morphology, Gram stain and a battery of biochemical tests [17].

A modified form of the Kirby Bauer method was used to determine antibiotic susceptibility of bacterial isolates [18]. The antibiotics tested included ampicillin, amikacin, augmentin, ceftriaxone, ciprofloxacin, ceftadizime, chloramphenicol, cefuroxime, cefotaxime, cotrimoxazole, gentamicin, levofloxacin, meropenem, nitrofuratoin, norfloxacin, nalidixic acid, piperacillin and tetracycline (Oxoid Ltd., Basingstoke, UK). The antibiotic susceptibility testing procedure employed is briefly described as follows. The test organism was purified and emulsified in peptone water until the turbidity was comparable with 0.5% McFarland’s standard. A loopful of the suspension was transferred onto a Mueller-Hinton agar plate, and then a sterile cotton swab was used to streak the entire surface of the plate. Sterile forceps were used to apply the antibiotic discs to the surface of the agar plate and incubated at 37 °C for 18–24 h. Zone diameters around the antibiotic discs were measured and classified as sensitive or resistant based on the Clinical Laboratory Standard Institute (CLSI) break point system.

### 2.4. Data Analysis

All data collected were entered into Microsoft Excel and further analysed using STATA version 12 (Strata Corp, College Station, TX, USA). Univariable associations were performed between bacteriuria and the other study variables. Analysis of variance was used for numeric variables, whereas chi-square test was used for categorical variables. Subsequently, variables significantly associated with bacteriuria were used as independent variables in a logistic regression analysis to identify determinants of bacteriuria. The variables included in the data analysis were gender, age, marital status, stroke severity, pyuria, frequency of stroke, side of the body affected, hypertension and diabetes. Significance of the independent variables was assessed by *p* values and odds ratios from Wald statistics; *p* values < 0.05 were regarded as significant.

Prevalence of bacteriuria was calculated as the proportion of bacteriuria cases at enrolment of the stroke patients into the study and during the entire period of follow-up (period prevalence). Incidence of bacteriuria was calculated as the proportion of new bacteriuria cases during follow-up of the stroke patients. Analyses were also done to determine the proportion of ASB that converted to UTI and identify any changes in the microbial etiology of bacteriuria among the study participants.

### 2.5. Ethics Statement

The study was approved by the by the Ethical and Protocol Review Committee of the College of Health Sciences University of Ghana (MS-Et/M.6-P3.3/2014/2015), and informed consent was obtained from the study participants.

## 3. Results

### 3.1. Demographic and Clinical Features of the Study Participants

Demographic features of the 71 stroke patients recruited in the study are summarised in Table 1. The gender distribution, marital status, and highest education of the stroke outpatients and inpatients were similar. Overall, majority of the study participants were males (67.6%), married (70.4%) and the most common education attained was tertiary (39.4%). The overall mean age of the stroke patients was 57.9 ± 12.01 years; mean age of outpatients and inpatients were 58.7 ± 8.1 and 55.3 ± 13.9 respectively.

In terms of stroke type, ischemic stroke constituted 49.3% (35) of total stroke cases, while haemorrhagic stroke accounted for 16.9% (12); 32.4% (32) of the subjects did not have their stroke type classified on records. In 40.8% (29) of the stroke cases, the disease had affected the left half of the body, while for 42.3% (30) the right half of the body was affected; 2.8% (2) of the study participants had both sides of their body affected. Based on mRs scores, 82.4% (56) of the stroke patients had mild stroke while 17.6% (12) had severe stroke. A proportion of 94.2% (65) of the stroke cases were acute first time attacks while 5.8% (4) were recurrent strokes. Hypertension (81.7%) was the most prevalent risk factor among the stroke patients, followed by diabetes (23.9%). As shown in Table 2, the pattern of stroke type, side of the body affected, stroke severity and risk factors were similar among the stroke outpatients and inpatients. The time for hospital stay among the stroke inpatients ranged from 3 to 15 days with a mean of 9 days.

The clinical signs of urinary infections observed among the stroke patients were pyuria, haematuria, frequent urination and dysuria. Pyuria occurred in 6 outpatients and 3 inpatients, haematuria occurred in 6 outpatients and 1 inpatient, frequent urination occurred in 7 outpatients, while dysuria occurred in 2 outpatients.

### 3.2. Bacteriuria and Associated Risk Factors

The overall prevalence and incidence of bacteriuria among the study participants were 12.7% (9/71) and 2.8% (2/71) respectively. Prevalence of bacteriuria among stroke outpatients and inpatients were 10.9% (6/55) and 18.8% (3/16) respectively (*p* = 0.411). Incidence of bacteriuria among stroke outpatients and inpatients were 1.8% (1/55) and 6.3% (1/16) respectively. Overall, 1/9 (11%) of the bacteriuria cases among the stroke patients was symptomatic (UTI) and this occurred in an outpatient who at the time of recruitment had ASB; there were no symptomatic cases among the inpatients. There were 13 cases where the patients had urinary symptoms but urine cultures were negative, and were therefore not classified as UTI.

The univariable analysis showed that none of the demographic features of the study participants including age, sex, marital status and education was significantly associated with bacteriuria. The only stroke parameter significantly associated with UTI in the univariable analysis was stroke severity (*p* < 0.001); stroke frequency, duration, subtype, side of the body affected and risk factors did not affect bacteriuria. Pyuria was significantly associated with bacteriuria in the univariable analysis (*p* = 0.001). In the logistic regression analysis, severe stroke (OR = 17.7, *p* = 0.008) and pyuria (OR = 38.7, *p* = 0.002) were identified as predictors of bacteriuria.

### 3.3. Uropathogens and Antibiotic Susceptibility

Nine bacteria were isolated from the urine specimens collected from the study participants (Table 3). *E. coli* was the most common bacterial organism isolated (33.3%) followed by *Staphylococcus* spp. (22.2%). The aetiological agents isolated from stroke outpatients appeared to be more diverse than those isolated from stroke inpatients (Table 3). Antibiotic susceptibility testing was carried out on *E. coli*, the main uropathogen isolated (Table 4). All the *E. coli* isolates were susceptible to amikacin, but resistant to augmentin, ampicillin, cefuroxime, cotrimoxazole, meropenem, norfloxacin and tetracycline.

## 4. Discussion

In this study we investigated the epidemiology of post stroke bacteriuria among stroke patients at KBTH in Ghana. A large majority of the stroke outpatients and inpatients were males, an observation which concurs with previous stroke studies in Ghana [5], indicating that stroke may be relatively more common among males in the country. The higher prevalence of bacteriuria among stroke inpatients (18.8%) compared with stroke outpatients (10.9%) concurs with a study carried out in Nigeria [19]. In the Nigerian study, the prevalence of bacteriuria among stroke inpatients and outpatients were 12.3% and 9.3% respectively [19]. Higher prevalence/incidence of bacteriuria among the stroke inpatients compared to the stroke outpatients, indicate that hospital acquired bacteriuria may be a more common complication of stroke than community-acquired bacteriuria. Maximal neurological impairment in stroke occurs during the first three days of the disease [20], by which time the patient is normally on admission in the hospital. Immunodepression resulting from the neurological impairment coupled with the invasive hospital procedures that stroke patients are usually subjected to, may account for the higher prevalence of bacteriuria among inpatients compared to outpatients. In the current study, stroke inpatients were hospitalised for an average of nine days and received invasive management procedures such as insertion of intravenous lines for medications and nasogastric tubes for feeding, though none of them were catheterised.

The association of stroke severity with bacteriuria concurs with previous studies [21,22]. Severe stroke patients are likely to receive more invasive management, increasing their risk of infections. Pyuria is one of the criteria for diagnosis of UTI [23,24], and therefore its association with bacteriuria in this study is not surprising. Increased white blood cells in urine (pyuria) may be an indication of urinary tract inflammation that could be an evidence of an infection, breach of mucosal lining and response to structural abnormalities [25]. Inflammation and mucosal breach predisposes the human host to colonization with uropathogens [26]. Diabetes has been reported to be significantly associated with UTI among stroke patients [10], though we did not observe this relationship; in the statistical analysis, the *p*-value (0.06) was almost significant. Diabetics may present with glycosuria, which supports the proliferation of uropathogens implicated in urinary infections [27]. Diabetes mellitus may also increase the risk of patients developing UTI via the impairment of the immune system as a result of hyperglycaemia and diabetes associated microangiopathy [28]. We did not find any signficant association between bacteriuria and several stroke parameters (subtype, side of the body affected, recurrence and hypertension) which has been previously reported [14].

In line with several studies on ASB and UTI [9,28], staphylococci and *E. coli* were the predominant aetiological agents among the stroke patients. In Ghana, a study by Donkor et al. [14] showed that the aetiology of community acquired bacteriuria UTI in stroke patients and healthy controls were similar and coagulase-negative staphylococci were the most prevalent in both populations. In the current study, *E. coli* resistance to ampicillin, norfloxacin and cotrimoxazole, which are potential antibiotics for treatment of UTI, is worth noting. Some of these antibiotics like ampicillin and cotrimoxazole have been on the Ghanaian market for a long time and there is evidence of their misuse resulting in high of level of resistance, which has been reported in other studies [29,30].

The majority of the bacteriuria cases among the stroke patients were asymptomatic, which concurs with the study in Ghana by Donkor et al. [14]. It is unknown if ASB among stroke patients actually progresses to symptomatic UTI, as there are hardly any longitudinal studies on the subject. Our data seems to provide evidence of the progression of ASB to symptomatic UTI in stroke patients, but this should be interpreted with caution given the small size. Further studies with larger sample size and longer periods of follow-up are needed to elucidate this relationship between ASB and symptomatic UTI among stroke patients. Screening for ASB among stroke patients is rare at both hospital and community levels. However, the high prevalence of ASB in this study coupled with the evidence of its progression to symptomatic UTI in stroke patients seems to indicate that screening for ASB among stroke patients may be necessary, as well as its treatment.

There are a few limitations of the study. Firstly, it was difficult recruiting stroke inpatients compared to outpatients. Consequently, we obtained an uneven distribution of the two groups of study subjects making their comparison a bit problematic. Secondly, while the cases of bacteriuria incidence in this study can be described as post-stroke infections, this may not be the situation for the prevalence cases. It is possible that some of the bacteriuria cases used in computing prevalence occurred prior to stroke in the patients, though this is unlikely considering the sampling strategy used.

## 5. Conclusions

Bacteriuria is a common complication among both stroke inpatients and outpatients at KBTH, though it appears to be more common among the former. Stroke severity appears to be the main stroke related determinant of bacteriuria among stroke patients. Bacteriuria among stroke patients is mainly asymptomatic and *E. coli* is the most important aetiological agent.

## Figures and Tables

**Table 1 medsci-05-00011-t001:** Demographic features of the study participants.

Feature	Outpatients		Inpatients		Overall	
	*n*	%	*n*	%	*n*	%
Gender						
Male	36	65.5	12	75.0	48	67.6
Female	19	34.5	4	25.0	23	32.4
Marital Status						
Single	3	5.5	2	12.5	5	7.0
Married	37	67.3	13	81.3	50	70.4
Separated/Divorced	3	5.5	0	0.0	3	4.2
Widowed	8	14.5	1	6.3	9	12.7
Education						
Primary	13	23.6	2	12.5	15	21.1
Secondary	9	16.4	4	25.0	13	18.3
Tertiary	20	36.4	8	50.0	28	39.4
None	4	7.3	2	12.5	6	8.5

Mean age of stroke patients was 57.9 ± 12.01 years (overall); 58.7 ± 8.1 years (out-patients) and 55.3 ± 13.9 years (in-patients).

**Table 2 medsci-05-00011-t002:** Stroke related clinical features of the study participants.

Feature	Outpatients		Inpatients		Overall	
	*n*	%	*n*	%	*n*	%
Stroke Subtype						
Ischaemic	25	45.5	10	62.5	35	49.3
Haemorrhagic	6	10.9	6	37.5	12	16.9
Data not available	23	41.8	0	0.0	23	32.4
Side of Body Affected by Stroke						
Right	25	45.5	5	31.3	30	42.3
Left	23	41.8	6	37.5	29	40.8
Left and right	2	3.6	0	0.0	2	2.8
Stroke Severity						
Mild stroke	47	90.4	9	56.3	56	82.4
Severe stroke	5	9.6	7	43.7	12	17.6
Frequency of Stroke						
One episode	51	96.2	14	87.5	65	94.2
Multiple episodes	2	3.8	2	12.5	4	5.8
Stroke Risk Factors						
Hypertension	46	83.6	12	75.0	58	81.7
Diabetes	16	29.1	1	6.3	17	23.9
Family History	1	1.8	1	6.3	2	2.8
Dyslipidaemia	4	7.3	1	6.3	5	7.0
Atrial fibrillation	1	1.8	1	6.3	2	2.8
Others	1	1.8	2	12.5	3	4.2

**Table 3 medsci-05-00011-t003:** Bacteriuria prevalence, incidence and causative organisms among stroke patients.

Parameter	Outpatients		Inpatients		Overall	
	*n*	%	*n*	%	*n*	%
Prevalence of bacteriuria	6	10.9	3	18.8	9	12.7
Incidence of bacteriuria	1	1.8	1	6.3	2	2.8
Causative organisms						
*Escherichia coli*	2	3.6	1	6.3	3	4.2
*Coagulase negative staphylococcus*	1	1.8	1	6.3	2	2.8
*Moraxella catarrhalis*	1	1.8	0	0	1	1.4
*Moraxella morgani*	0	0	1	6.3	1	1.4
*Klebsiella pneumoniae*	1	1.8	0	0	1	1.4
*Enterococcus faecalis*	1	1.8	0	0	1	1.4

**Table 4 medsci-05-00011-t004:** Antibiogram of *Escherichia coli* isolates from urine specimens of stroke patients.

Antibiotics	Isolate 1	Isolate 2	Isolate 3
Ceftriaxone	R	R	S
Ciprofloxacin	S	R	R
Augmentin	R	R	R
Ceftadizime	I	R	S
Nalidixic acid	I	R	R
Norfloxacin	R	R	R
Levofloxacin	S	R	R
Piperacillin	S	R	S
Gentamycin	S	R	S
Amikacin	S	S	S
Nitrofuratoin	S	I	R
Tetracycline	R	R	R
Cefuroxime	R	R	R
Chloramphenicol	S	S	S
Cotrimoxazole	R	R	R
Ampicillin	R	R	R
Cefotaxime	R	R	S
Meropenem	R	R	R

R: resistant; S: susceptible; I: intermediate resistance.

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
