# Peer review of "Post-Stroke Bacteriuria: A Longitudinal Study among Stroke Outpatients and Inpatients at the Korle-Bu Teaching Hospital in Ghana"

_medsci, 2017, doi:10.3390/medsci5020011_

Reviewer 1 Report

This study talks about the incidence of UTI in stroke patients in Ghana. I think the biggest drawback of this study is that it is very difficult to draw any conclusion because of the uneven distribution and the low patient numbers. 

There are only 16 in-patients while there are 55 out-patients. With such an uneven distribution, it is extremely hard to make conclusions in the findings in the outpatients vs the outpatients. Another Question I had was what was the age of the patients in the study (especially the male patients). We have to eliminate the possibility of prostate enlargement in the older male patients if the patients.

Author Response

We agree with the reviewer about the uneven distribution of stroke outpatients and inpatients. This occurred because we had difficulty in recruiting outpatients. We have indicated this point as a limitation of the study (please see lines 491-493; last paragraph of discussion). This point affects only one of the three conclusions of the study ie the first conclusion, which attempts to compares bacteriuria in outpatients and inpatients.

The age of the study participants is indicated in the footnote of Table 1. The ages of the males and females were similar; we have presented mean ages of outpatients and inpatients for purposes of uniformity in the data presentation.

None of the male patients had prostate enlargement.

Reviewer 2 Report

Background

It is a very ambitious study however with some weaknesses which is needed to solve out.

The definition of UTI ? – in this case the authors present UTI as similar with asymptomatic bacteriuria. What this study actually is presenting is the prevalence of bacteriuria (ASB)– not UTI – this have to be distinglished! See reference (Bjerklund Johansen T., Botto H., Cek M.,  et al 2011)

The prevalence rates of asymptomatic bacteriuria has to be presented according to other prevalence studies of a normal population – and according to them is not uncommon with ABU in male and females over 50 years old, and increase with age.

Methods

When prevalence rate of UTI is presented. From what urine culture is that classified. It is written at the enrolment of the study. Is that the same time as admitted to the hospital ? The title tells us UTI after stoke, How long time after ? The presentation of incidence is more clear.

In the method it would have been mentioned that no one of the stroke patients had been cauterized and had an in dwelling catheter. It is mentioned in the discussion, but it would be marked in the method or results.

 Why did no one of the patients did had any urinary problems. Urinary incontinence and urinary retention is one of the remakes of stroke severity. How come that nobody of your population had any of these problems? Did you measure residual urine?

Urinary incontinence is very common after stroke. How was the situation among these patients? (ref. Brittain, Peet, & Casteden 1998; Kovinda, Wattanapan, Permsirivanich & Kuptniratsaikul 2009)

It is not mentioned how many of the patients received any antibiotic treatment.

What is the difference in the outpatient and inpatient groups – it is not clarified

We lack any information of the functional status of the informants, there ability to mobilize, communicate etc.  and if any of these factors were entered into the logistic analyze.

Statistics and Results

It is not mentioned how many and witch variables was included in the logistic analysis. The total among of responders were around 70 and there would be a limit of 7 variables included in this analysis due to the literature. Please explain which variables were included !

Such variables as pyuria seems to be not necessary to include (as it is a obvious symptom of UTI) and thus would be replaced with another.

It is interesting finding that African stroke patients have lower incidence of asymptomatic bacteriuria compared to European countries. It would be interesting if you would lift this in the discussion part. Why do not refer to any other references than African studies. For instance (Stenzelius, Laszlo, Madeja & Grabe, 2016) were we investigated symptomatic UTI in stroke patients with urinary catheters. That study showed of course much higher symptomatic UTI then you have presented.

The difference in sexes in the outpatient and inpatient groups were very different. Why were it important to differ between this two groups?

Were gender a variable in the logistic regression model? This is important to mention as UTI is quite more common among females compare to men.

Author Response

BACKGROUND:

1      The definition of UTI has been redefined in the manuscript with appropriate reference. We have also distinguished between UTI and ASB (Please refer to line number 89-93; 2nd paragraph of Introduction).

2      Reference to prevalence rate of ABU and UTI in healthy populations has been included in the introduction. Please refer to line number 95-97; 2nd paragraph of Introduction

METHODS:

1.     Prevalence was calculated based on the first urine specimens collected from the patients at enrollment. For out-patients, this urine specimen was collected on the first day of visit to the physiotherapy clinic of the study hospital; for in-patients, it was the first urine specimen collected on admission (see lines 322-325; 2nd paragraph of data analysis). We do not have information on the exact period of the stroke prior to determining bacteriuria prevalence. However, for outpatients, the study participants had had stroke for at least 3 months (please see lines 246-247; 1st paragraph under Study design, recruitment and data collection). We have discussed this limitation in the discussion section (lines 480-484; last paragraph).

2.     It has been mentioned in the Methods Section of the manuscript that none of the stroke patients was catheterized (please see line 265-266; last paragraph of Study design, recruitment and data collection).

3.     Residual urine was not measured. We did not investigate urinary incontinence and urinary retention. Most stroke patients seen at the physiotherapy clinic were recovered from stroke and were taking physiotherapy  routines to address Post-Stroke related physical disabilities of various degrees. As such most of them did not exhibit severe urinary incontinence. With hospitalized participants, incontinent patients were not included in the study since sampling would be difficult without an invasive approach.

4.     None of the study participants received antibiotics from the study except if it was in line with clinical management of the patient. The Cohort ended for any participant who had to receive antibiotic treatment in the course of clinical management. Please see lines 266-268; last paragraph of Study design, recruitment and data collection).

5.        Difference between in- patient and out- patient groups have been clarified in line 242-247 (first paragraph of Study design, recruitment and data collection).

6.     Functional status of the patients was captured by the Modified Rankin Score (mRs), which assesses severity of stroke by functional ability variables (please see lines 257-259; 2nd paragraph under Study design, recruitment and data collection).

STATISTICS AND RESULTS:

1      The variables included in the in the logistic regression analysis are listed in line 331-332 (1st paragraph of data analysis)

2      The authors think that it is not out of place to include pyuria in the regression analysis. The significant association of pyuria with bacteriuria in the study adds to the body of evidence of pyuria being an important diagnostic criterion for bacteriuria.

3      Incidence in our study was lower compared to other studies in Europe as cited in the review comments. However, these studies investigated UTI in catheterized patients who have a greater risk of developing UTI. Indwelling catheters have been shown to increase risk if UTI. Our population were not catheterized and thus we could not compare our findings to such studies.

4      Among both outpatients and inpatients, males were predominant and we have commented on this (please see lines 426-429; 1st paragraph of discussion). The difference between male inpatients and outpatients was about 10%. Similarly, the difference between female inpatients and outpatients was about 10%. We think these differences are not so high and may have happened by chance.

5      Yes gender was included in in the logistic regression model (Please see lines 331-332; 1st paragraph of data analysis).

Round  2

Reviewer 1 Report

Thanks for addressing my comments. No other comments from my side

Author Response

We are grateful to the reviewer for the useful suggestions to improve the paper.

Reviewer 2 Report

The manuscript has improved bet there are still some corrections/ explanations to be done.

The title is about bacterieuria – but the introduction ends with another aim of the study.

The study is longitudinal – but most data analyzed is only cross-sectional, except for the calculations of incidence. It is however not comparable to talk about incidence in inpatients during hospital time – how many days did they spend in the ward? The outpatients incidence is calculated in a period of 6 months. – to my mind this is not interesting and can not be summarized as it is in the abstract – presented in percent and p-values as the sample size is too small 1/55 or 1/16

The prevalence of bacterieuria in the inpatient group is measured based on the urine sample taken at admission to the ward. This is rather a measure of bacteriuria among elderly in a population.

Too much attention is taken in the abstract and discussion about the evidence that ASB progress unto UTI. This study can not claim to prove any evidence as the sample size is too small. Please, be more humble.

Reference list: Why referre to Shaik et al 2016 and Roberts et al 2011 – studies among pediatrics ?

Reference 24 is not complete

Reference Roberts et al 2011 have no number.

Author Response

We thank the reviewer for the useful comments made. Our responses to the comments are provided below and the corresponding modifications in the manuscript are in green colour.

We have corrected the study aim to read “to investigate the epidemiology of post-stroke bacteriuria among stroke patients at the Korle-Bu Teaching Hospital in Ghana, including the prevalence, incidence, risk factors and aetiological agents”. (Please see the last paragraph of the Introduction).

The study is a longitudinal one because we followed up the stroke patients. Apart from calculation of incidence, we also analysed the data to determine the proportion of asymptomatic bacteriuria that converted to symptomatic bacteriuria (Urinary Tract Infection). Additionally, we analysed that data to identify changes in the microbial etiology of bacteriuria among the study participants (please see data analysis section).

In the abstract, we have corrected the problem of comparing incidence of bacteriuria in oupatients and inpatients. We have rephrased the statement concerned to read “Among both the outpatients and inpatients there was one new case of bacteriuria each during the period of follow up”. Please see the results section of the abstract.

Previously, we measured bacteriuria prevalence based on the urine samples taken at enrolment (point prevalence). In the current revised manuscript, we have rather presented period prevalence as this provides a better picture of the study (please see data analysis section).

We have toned down our argument of ASB progressing to UTI. We have rephrase the statement concerned to read “Our data seems to provide evidence of the progression of ASB to symptomatic UTI in stroke patients, but this should be interpreted with caution given the small size. Further studies with larger sample size and longer periods of follow up are needed to elucidate this relationship between ASB and symptomatic UTI among stroke patients. Please see the 4th paragraph of the discussion section. We have also avoided the point ASB progressing to UTI in the abstract.

References 24 and 25 are used in the context of diagnosis of bacteriuria and UTI in relation to pyuria. This is general and not necessarily stroke related. We have however replaced reference 25 so that the evidence showing the relationship between bacteriuria and pyuria is not limited to only children. We have also addressed all the comments about references.

Round  3

Reviewer 2 Report

The manuscript is very much improved and you have taken all my suggestions and changed the manuscript. and to my mind it can be published now.